# Rare missense variants in the human cytosolic antibody receptor preserve antiviral function

**Jingwei Zeng[†], Greg Slodkowicz[†], Leo C James\***

MRC Laboratory of Molecular Biology, Cambridge, United Kingdom

**Abstract** The genetic basis of most human disease cannot be explained by common variants. One solution to this 'missing heritability problem' may be rare missense variants, which are individually scarce but collectively abundant. However, the phenotypic impact of rare variants is under-appreciated as gene function is normally studied in the context of a single 'wild-type' sequence. Here, we explore the impact of naturally occurring missense variants in the human population on the cytosolic antibody receptor TRIM21, using volunteer cells with variant haplotypes, CRISPR gene editing and functional reconstitution. In combination with data from a panel of computational predictors, the results suggest that protein robustness and purifying selection ensure that function is remarkably well-maintained despite coding variation.
DOI: https://doi.org/10.7554/eLife.48339.001

## Introduction

Rare missense variants outnumber common ones, with 85% of non-synonymous variants displaying a minor allele frequency of less than 0.5% (*Abecasis et al., 2012*), and 200–300 such alleles per sequenced individual (*Bamshad et al., 2011*). As each specific variant is present at a very low frequency within the population, the impact on human health is hard to assess. Collectively however, rare variants are thought to be a significant component of the 'missing heritability' paradigm and their neglected contribution may explain why only a fraction of inherited diseases are genetically accounted for *Maher (2008)*. Classic GWAS approaches lack the power to correlate trait heritability with rare coding alleles (*Auer and Lettre, 2015*; *Bomba et al., 2017*); indeed, they are largely limited to the identification of common variants with small effect sizes, particularly those within regulatory regions (*Astle et al., 2016*). Yet, the clear inverse correlation between allele frequency and trait impact suggests that rare variants are more likely to be disease-causing (*Kryukov et al., 2007*; *Park et al., 2011*). Common variants have undergone purifying selection and are therefore more likely to be benign. In contrast, a higher proportion of rare variants will be functionally damaging as they mostly comprise recent or *de novo* mutations (*Keinan and Clark, 2012*) on which selection has not yet acted. Multiple different rare *de novo* mutations are thought to underlie the genetics of many complex human disorders including schizophrenia, epilepsy, lipid metabolism disorder, and inflammatory disease (*Andrews et al., 2013*; *McClellan and King, 2010*). Estimates from the 1000 Genomes Project suggest that 40% of rare missense mutations are damaging compared to 5% of common variants (*Abecasis et al., 2010*).

While the advent of next-generation sequencing (NGS) has made obtaining human sequence data straightforward and inexpensive, linking genotype to phenotype is far less trivial. Sophisticated computational tools have been produced in order to predict the functional impact of missense variants. Early prediction methods typically utilized a combination of sequence conservation and amino-acid properties while newer tools typically employ ensemble methods that integrate a large number of diverse features using machine learning. Unfortunately, these predictions are not always

**\*For correspondence:**
lcj@mrc-lmb.cam.ac.uk

[†]These authors contributed equally to this work

**Competing interests:** The authors declare that no competing interests exist.

prognostic of disease severity or outcome. A study of the cystic fibrosis gene CFTR found a poor correlation between predicted functional impact and disease (*Dorfman et al., 2010*), while in silico classification of rare BRCA1/2 mutations was not predictive of pathogenicity (*Ernst et al., 2018*). A direct assessment of multiple computational methods, carried out as part of the Critical Assessment of Genome Interpretation, compared phenotypic predictions with an empirical dataset quantifying the ability of SUMO-conjugating enzyme UBE2I variants to rescue the growth of *S. cerevisiae* (*Zhang et al., 2017*). While most predictors identified deleterious mutations at conserved positions, they were less accurate at non-conserved sites and gave false-positive predictions.

The most direct way to assess the phenotypic contribution of a missense mutation is through empirical testing. A landmark study introduced 33 *de novo* missense mutations by random mutagenesis into immune genes and measured the impact on lymphocyte subsets in homozygous mice (*Miosge et al., 2015*). Strikingly, only 20% of variants predicted by computational methods to be deleterious gave an observable phenotype. The same study found that while approximately 50% of *de novo* missense mutations within the same species were predicted to be functionally impaired, this compared with only 5% of the variants found between-species. This would suggest that many variants possess near neutral phenotypes not discernible *in vivo* yet sufficiently impactful to undergo purifying selection. Moreover, it highlights a crucial general question: are predicted deleterious mutations actually often neutral or does phenotypic characterization fail to capture their impact?

We decided to address this question by investigating how naturally occurring variants impact the cytosolic antibody receptor TRIM21, using multiple molecular and cellular assays that independently quantify protein stability, function and phenotype. TRIM21 intercepts incoming antibody-coated pathogens during cellular infection and causes them to be degraded by the proteasome. TRIM21 also activates immune signaling pathways, including NF-κB, although this is tightly regulated to prevent inopportune inflammation. These disparate complex functions are achieved using multiple component domains and by recruiting a range of cofactors. We determined the protein stability, ligand binding, viral neutralization, innate immune signaling, protein expression and dominant-negative potential of rare missense variants in TRIM21 identified from the gnomAD consortium and 1000 Genomes Project (*Lek et al., 2016*; *Sudmant et al., 2015*). We characterized variant activity in cells from 1000 Genomes volunteers, by CRISPR gene editing and through a novel reconstitution system that recapitulates endogenous expression and regulation.

## Results

To provide a comprehensive description of natural variation in *TRIM21*, we considered all missense variants from gnomAD, which comprises 125,748 exome sequences and 15,708 genome sequences, including data from the 1000 Genomes Project. Analysis of these variation data revealed a total of 277 rare missense variants, defined as having less than 1% minor allele frequency (MAF) and no common variants (≥1% MAF). Of these rare variants, the highest allele frequency is 0.32% for V50I, with 882 heterozygotes and nine homozygotes. We compared non-synonymous genetic variability within the *TRIM21* gene to related TRIMs and gene families with high (*HDAC* and *actin* genes) and low (*HLAs*) levels of conservation. Degree (frequency) of variation was quantified as the number of missense variants per 100 amino-acid positions (AAs) to take account of differing gene size (*Figure 1*). There was a substantial difference in the frequency of common variants between genes, which were more prevalent in immune genes *STING*, *TRIM20* and *TRIM5* than in *TRIM21* (*Figure 1A*). Indeed, *TRIM21*, is more invariant than *HDAC* and similar to actin in its fixation of a common allele. Both *STING* (*Mozzi et al., 2015*; *Yi et al., 2013*) and *TRIM5* (*Sawyer et al., 2005*) are thought to be under positive selection, as evidenced by their high dN/dS in interspecies sequence comparisons. In human populations, variants in these genes display geographic variation, suggesting that exposure to pathogens explains the increased frequency of common variants associated with these genes (*Figure 1A*). TRIM20 is associated with familial Mediterranean fever and the existence of common deleterious geographic variants is thought to be a result of past heterozygous advantage (*Manukyan and Aminov, 2016*). In contrast, TRIM21 has no common missense variants, consistent with purifying selection. This difference between TRIM5 and TRIM21, despite both detecting incoming viruses and restricting replication, is consistent with their different binding mechanisms. TRIM5 binds directly to retroviral capsids, which are continually evolving, whereas TRIM21 binds viral capsids via attached antibodies, which are conserved within their Fc region. There was noticeably

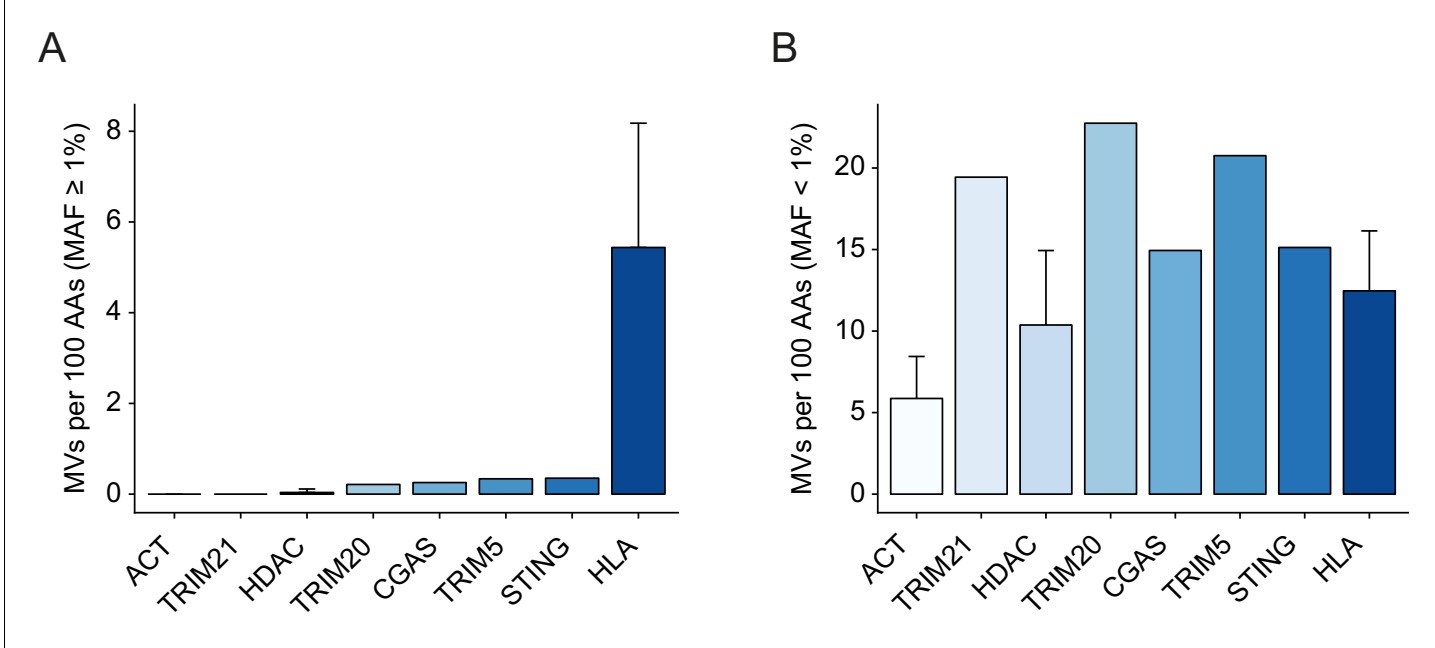

**Figure 1.** *TRIM21* is a highly conserved gene. Conservation of *TRIM21* compared to invariant housekeeping genes (*HDACs* and *actins*), and immune genes including *CGAS, STING* and related TRIMs, *TRIM5* and *TRIM20*. Comparison of variants between genes is expressed as number of missense variants (MVs) per 100 residues to normalize for differences in gene length (A-B). Frequency of nonsynonymous variants (A) common variants (≥1%) and (B) rare variants (<1%). Error bars represent standard deviation. Data provided in *Figure 1—source data 1*.

DOI: https://doi.org/10.7554/eLife.48339.002

The following source data is available for figure 1:

**Source data 1.**

DOI: https://doi.org/10.7554/eLife.48339.003

less variation in the frequency of rare variants between genes (*Figure 1B*). This is consistent with rare human SNP frequencies being significantly impacted by *de novo* mutation rates (*Pritchard, 2001*).

Within the 1000 Genomes Project (2504 individuals), there are 22 rare variants in *TRIM21*, all of which are heterozygous, although for four of them, there are individuals in gnomAD who are homozygous (E38G, V50I, R57C and Q88K). We focused on the 1000 Genomes panel of variants because immortalized cell lines are available from these volunteers, allowing each haplotype to be tested in its naturally occurring genomic context. We first assessed whether identified variants are likely to perturb TRIM21 activity by utilizing a panel of phenotypic prediction algorithms, including PolyPhen-2 (*Adzhubei et al., 2010*), SIFT (*Sim et al., 2012*), CADD (*Rentzsch et al., 2019*), REVEL (*Ioannidis et al., 2016*), MetaLR (*Dong et al., 2015*) and MutationAssessor (*Reva et al., 2011*) (*Supplementary file 1*). Of the 22 variants, 21 are missense and one is a premature stop codon. PolyPhen-2 predicts 7/21 variants to be possibly or probably deleterious, SIFT 4/21, while CADD, REVEL and MetaLR predict only one variant, F446I, to be deleterious with the remaining either tolerated or benign (*Supplementary file 1*). Taken together, there are noticeably fewer predicted deleterious variants in TRIM21 than have been reported for other immune genes (*Miosge et al., 2015*) or for hominid genes in general (*Eyre-Walker and Keightley, 1999*). We hypothesized that some of this difference may be due to the increase in the number of genomes now available. To test this, in addition to SIFT and PolyPhen-2 predictions from the current version of Ensembl, we obtained equivalent predictions based on earlier versions of the underlying sequence databases (Materials and methods) and found that this caused both SIFT and PolyPhen-2 to predict a higher proportion of deleterious variants (8/21 and 9/21, respectively).

Next, we sought to empirically determine the impact of natural variants on TRIM21 function and assess the accuracy of phenotypic prediction. The 22 variants found in the 1000 Genomes Project

are distributed amongst all 4 TRIM21 domains – the RING, B-Box, coiled-coil and PRYSPRY. Structural data is available for all domains (albeit the coiled-coil is based on a model of TRIM25) and the location of each variant is marked (*Figure 2*).

We began by focussing on the PRYSPRY as this is the only globular domain in TRIM21 and is responsible for IgG binding. We expressed each of the seven variants and compared their binding of IgG Fc with wild-type by ITC (*Figure 3*, *Figure 3—figure supplement 1*). Wild-type PRYSPRY bound IgG with an affinity of 9.4 nM (*Figure 3B*). Most variants maintained binding affinity within error of wild-type, with K455E being the weakest at 12 nM and A390V and Q470K the strongest at 1.7 and 1.2 nM respectively (*Figure 3B*). These results are in contrast to predictions, where SIFT categorized 4 of the PRYSPRY variants as deleterious, PolyPhen-2 predicted three to be possibly damaging and the other algorithms only F446I.

Importantly, whilst binding was maintained for all variants, we noted significant changes in protein behavior, in particular with F446I. Most variants expressed with a yield of 2 mg/l and could be concentrated to 500 µM. In contrast, F446I gave yields of 10–100 µg/l and could not be concentrated to levels > 30 µM. G440R also had reduced expression at 0.1 mg/l (*Figure 3—figure supplement 2*). This suggests that while IgG binding was unaffected, PRYSPRY variants could impact function indirectly by altering protein stability. The PRYSPRY domain of TRIM21 is a β-sandwich with a classic globular fold. Several of the variants map to, or are in the vicinity of, the hydrophobic core. To directly determine the impact of each variant on PRYSPRY stability we measured the melting temperature (Tm) of all variants using a Prometheus differential scanning fluorimeter (DSF) (*Figure 4A*). As the Prometheus DSF uses natural tryptophan fluorescence rather than an extrinsic reporter dye, it avoids the potential issue of changes in dye binding. Measurement of wild-type PRYSPRY reveals it is a moderately stable protein with a Tm of ~48°C. We observed considerable variation in stability between naturally occurring PRYSPRY variants (*Figure 4B*), in contrast with their similarities in IgG binding. N297H displayed a slightly increased stability of 2°C, while A390V, G440R and F446I were all substantially less stable, consistent with their lower yield from *E. coli*.

To assess the accuracy of each prediction algorithm, we compared the empirically determined ΔΔG of natural PRYSPRY variants with the calculated prediction scores. Overall, there was remarkable correlation between observed and predicted effects, particularly for REVEL and MetaLR (*Figure 4C*). These programs therefore not only identified deleterious variants without calling false positives but also correctly ranked variants in terms of their relative stability. PolyPhen-2, while not the most accurate in this analysis, still nevertheless predicts as its potentially damaging variants those three with the lowest stability. Of all PRYSPRY variants tested, F446I was most unstable with a dramatic reduction in Tm of ~10°C. This result is consistent with the observed poor behavior of this protein in terms of bacterial expression and purification. Moreover, F446I is the most buried of all the variants, with its side chain packed close to the center of the core. Most importantly, this result is consistent with the identification of F446I by CADD, REVEL and MetaLR as the only deleterious TRIM21 variant (*Supplementary file 1*).

In contrast to the PRYSPRY, both the RING and B-Box domains of TRIM21 are not globular folds but Zn-fingers each of which is stabilized by two Zn ions (*Figure 2B*). This makes phenotypic prediction of variants in these domains difficult. However, both domains are essential to TRIM21 function. The RING domain of TRIM21 is a highly active E3 ubiquitin ligase but is tightly regulated to prevent constitutive ubiquitination and signaling down the NF-κB pathway. RING function is regulated by the B-Box domain, which is autoinhibitory and blocks E2 enzyme recruitment (*Dickson et al., 2018*). To investigate whether RING-Box variants have an impact on the ability of TRIM21 to activate NF-κB signaling in response to viral infection, we needed a system for expressing TRIM21 variants in a *TRIM21* KO cell line.

While plasmid transfection is a convenient method for over-expression of proteins in mammalian cell lines, viral promoters (e.g. CMV) typically induce expression that is orders of magnitude greater than endogenous levels, potentially masking subtle differences in activity, intrinsic stability and cellular turnover. We therefore sought a system in which we could ectopically express our panel of natural variants at endogenous levels and retain native transcriptional regulation. To this end, we constructed a lentiviral transduction vector utilizing 2 kb of endogenous *TRIM21* sequence predicted to contain native promoter elements (*Figure 5A*). We removed the endogenous copy of *TRIM21* from 293Ts by CRISPR-Cas9 and made stable cell lines of each variant by lentiviral transduction. Using our native transduction cassette, we could restore expression of TRIM21 back to endogenous

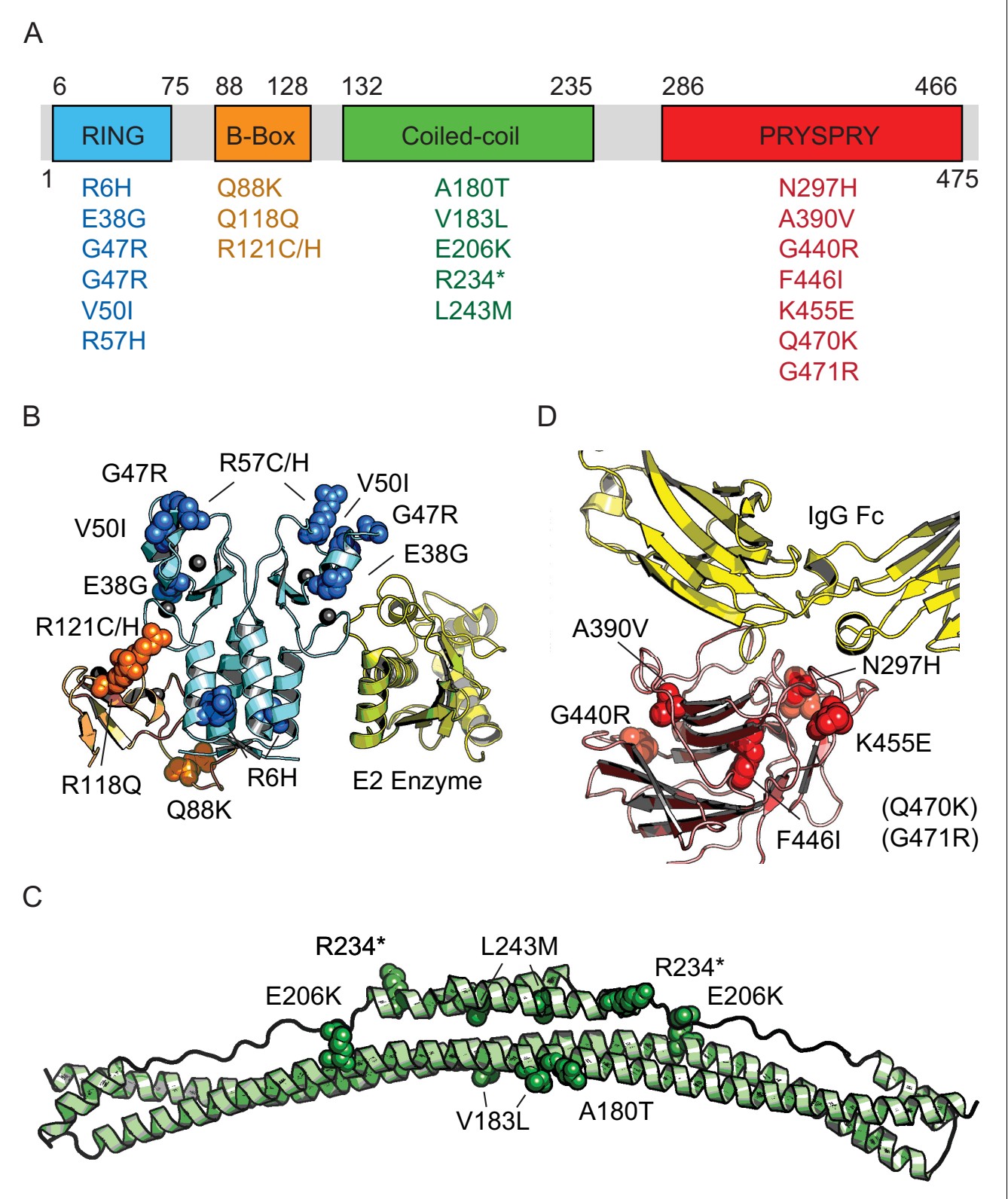

**Figure 2.** Domain location of TRIM21 missense variants. (**A**) Schematic of TRIM21 showing how variants are distributed amongst component domains. (**B–D**) Mapping of variants onto structures of each domain. (**B**) RING (blue) and B-Box (orange) variants are marked on the autoinhibited structure of the TRIM21 RING-B-Box domains (PDB 5OLM). The second copy of the B-Box has been omitted and the location of a bound E2 enzyme (yellow) has been included instead to indicate the location of this functional interface (based on superposition of TRIM25 RING: E2 structure (5FER)). (**C**) Coiled-coil

*Figure 2 continued on next page*

*Figure 2 continued*
domain (green) based on TRIM25 (PDB 4CFG). Location of TRIM21 residues are marked based on sequence alignment. (D) PRYSPRY domain (red) with bound IgG Fc (yellow) based on PDB 2IWG. Residues in brackets are not present in the structure.
DOI: https://doi.org/10.7554/eLife.48339.004

levels in the three different cell lines tested (*Figure 5C*, *Figure 5—figure supplement 1A–C*). The expression level was substantially lower than that driven by the SFFV promoter (*Figure 5C*). Moreover, reconstituted 293T cells upregulated TRIM21 in response to interferon-alpha, similar to unmodified parent cells (*Figure 5B*).

A specific problem with TRIM proteins is their propensity to spontaneously form higher-order structures when over-expressed (*Fletcher et al., 2018*; *Reymond et al., 2001*). Over-expression of TRIM21 has previously been shown to cause large rod-shaped structures unlike the diffuse distribution of endogenous protein (*Rhodes et al., 2002*). We therefore also tested whether our bespoke native-promoter driven system properly recapitulates the ability of TRIM21 to detect incoming antibody-coated virions during infection. Using an mCherry-TRIM21 fusion, we observed good colocalization with AdV5-IgG complexes (*Figure 5—figure supplement 2*). This colocalization was specific

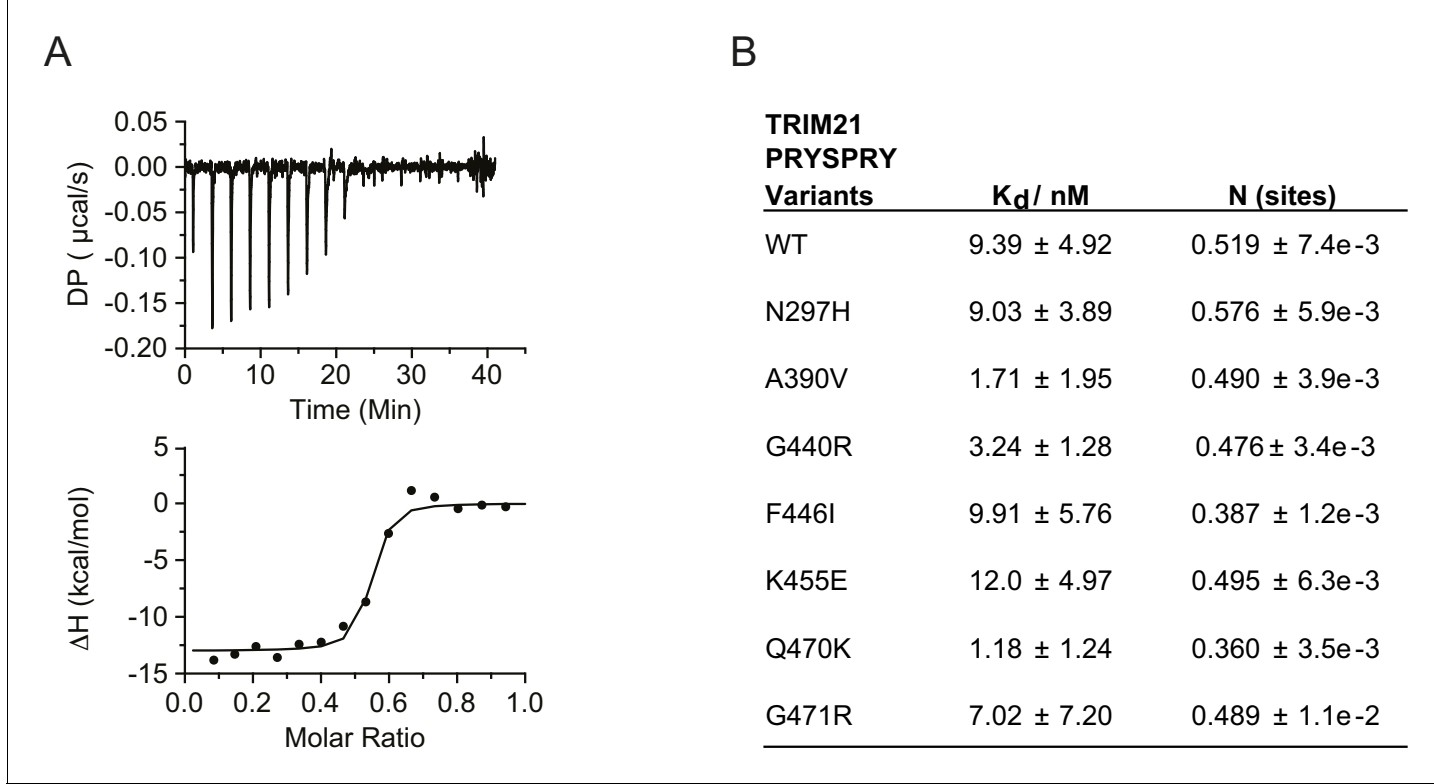

**Figure 3.** PRYSPRY missense variants maintain antibody binding. (A) Representative Isothermal titration calorimetry (ITC) trace of IgG Fc against WT TRIM21 PRYSPRY fitted to the one set of sites model. (B) Summary of PRYSPRY variant binding affinities to IgG Fc. Consistent with known binding mode, TRIM21 PRYSPRY binds IgG with a stoichiometry of 2:1 (*Keeble et al., 2008*). Data provided in *Figure 3—source data 1*.
DOI: https://doi.org/10.7554/eLife.48339.005

The following source data and figure supplements are available for figure 3:

**Source data 1.**
DOI: https://doi.org/10.7554/eLife.48339.008
**Figure supplement 1.** Representative ITC traces of IgG Fc titration against each PRYSPRY variant.
DOI: https://doi.org/10.7554/eLife.48339.006
**Figure supplement 2.** Purification of TRIM21 PRYSPRY protein.
DOI: https://doi.org/10.7554/eLife.48339.007

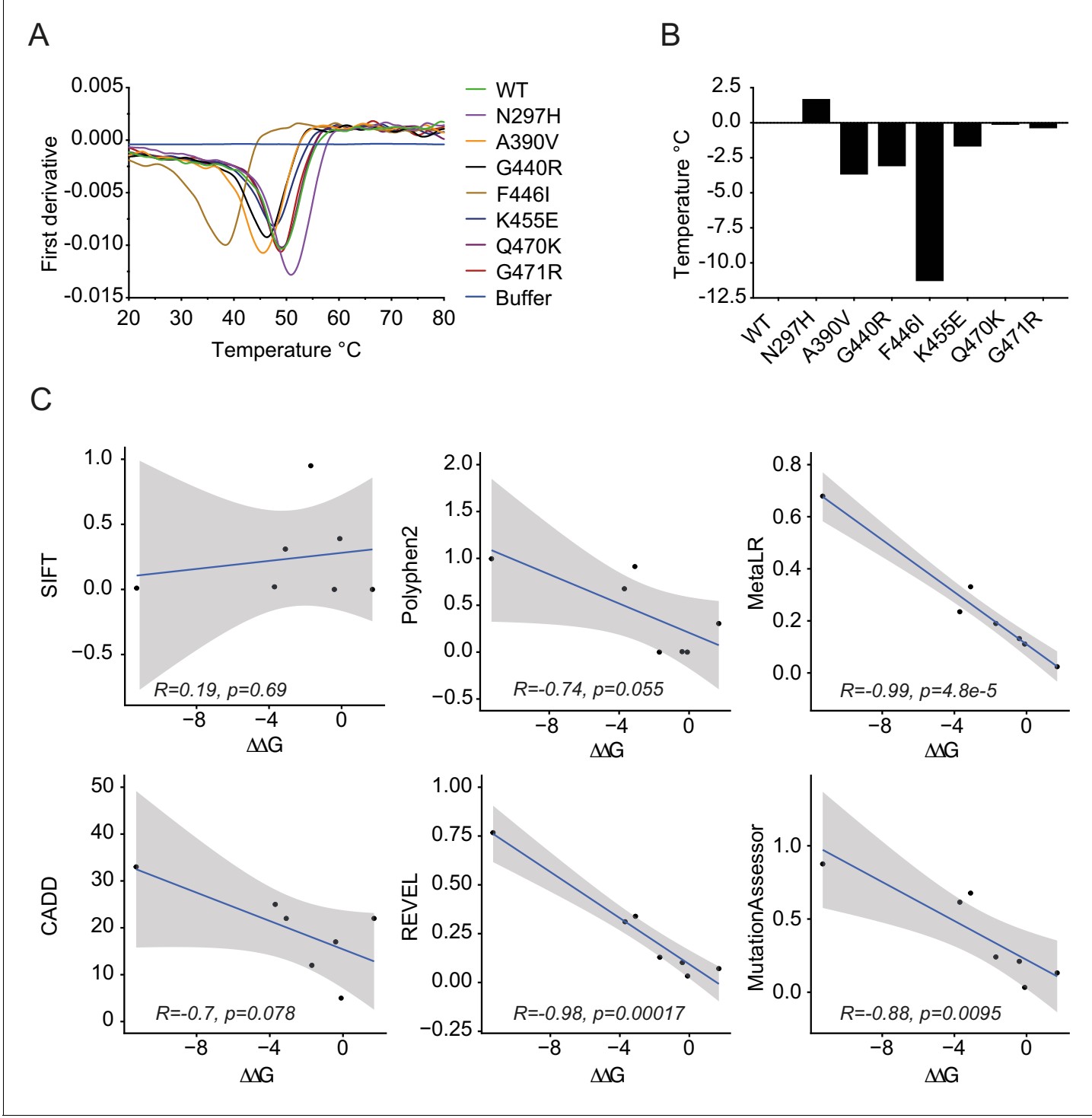

**Figure 4.** PRYSPRY variants have profound differences in intrinsic stability. (**A**) Differential scanning fluorimetry of PRYSPRY variants to determine their melting temperature (Tm) using changes in intrinsic tryptophan fluorescence that occur upon unfolding (first derivative of 330/350 nm ratio). (**B**) Differences between variants are plotted as a ΔTm with respective to wild-type. (**C**) Correlation between ΔΔG for stability of PRYSPRY variants with predicted degree of deleteriousness from various algorithms. Correlation and p-values were calculated using the Pearson correlation coefficient. Data provided in *Figure 4—source data 1*.

DOI: https://doi.org/10.7554/eLife.48339.009

The following source data is available for figure 4:

**Source data 1.**

DOI: https://doi.org/10.7554/eLife.48339.010

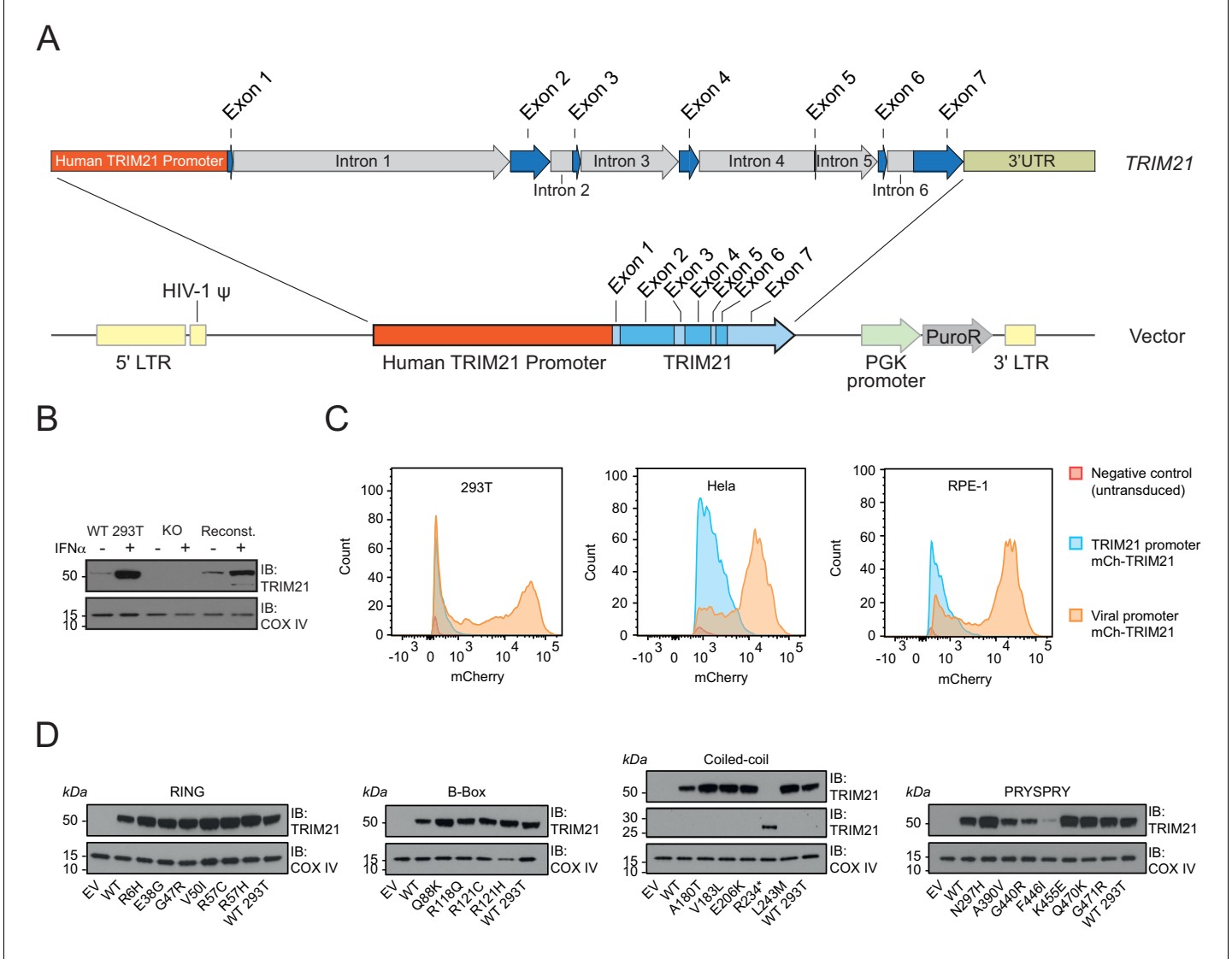

**Figure 5.** A bespoke system for ectopic TRIM21 expression at endogenous levels. (**A**) Map of lentivector containing the endogenous 2 kb upstream promoter sequence of the human *TRIM21* gene followed by the 5'UTR (Exons 1–2) and *TRIM21* coding sequence (Exons 2–7). (**B**) Immunoblot of TRIM21 and COX IV (loading control) in WT, TRIM21 KO (KO) or lentivector reconstituted (Reconst.) 293Ts with or without interferon-alpha (IFN-α) pre-treatment. (**C**) Histograms of mCherry fluorescence intensity in cells transduced with lentivector encoding mCherry-TRIM21 driven by SFFV (Viral; Orange) or native *TRIM21* promoter (Blue). Untransduced *TRIM21* KO 293Ts were used as negative control (Red). (**D**) Immunoblot of TRIM21 and COX IV (loading control) in lentivector reconstituted 239Ts expressing the indicated TRIM21 variant, with the variants grouped into their host domains.

DOI: https://doi.org/10.7554/eLife.48339.011

The following source data and figure supplements are available for figure 5:

**Source data 1.**
DOI: https://doi.org/10.7554/eLife.48339.014
**Figure supplement 1.** TRIM21 expression levels in reconstituted cell lines.
DOI: https://doi.org/10.7554/eLife.48339.012
**Figure supplement 2.** Native promoter driven mCherry-TRIM21 colocalizes with antibody coated AdV5.
DOI: https://doi.org/10.7554/eLife.48339.013

and dependent on the interaction between TRIM21 PRYSPRY and IgG Fc because the H433A mutation in IgG that abrogates TRIM21 binding also prevented colocalization. Comparing the TRIM21 expression levels in all 22 variant stable cell lines revealed that reconstitution was successful in most cases within twofold of wild-type (*Figure 5D*, *Figure 5—figure supplement 1C*). The obvious

outliers were A390V, G440R and F446I, which were substantially lower in expression. This result is consistent with the behaviour and intrinsic thermostability of these variants as recombinant proteins. Indeed, within the PRYSPRY variants there is a correlation ($R^2$ = 0.72) between expression levels and thermostability (*Figure 5—figure supplement 1E*).

During its effector response, TRIM21 neutralizes incoming virions with an efficiency that is proportional to antibody concentration. Previously, we have shown that neutralization is highly sensitive and that activity can be detected when a virion is coated with as few as two antibodies (*McEwan et al., 2012*). We carried out neutralization experiments on all 22 variants over a range of antibody concentrations (*Figure 6A*). Neutralization curves were fit to an exponential decay equation and a neutralization constant ($K_{neut}$) determined for each variant. Most variants

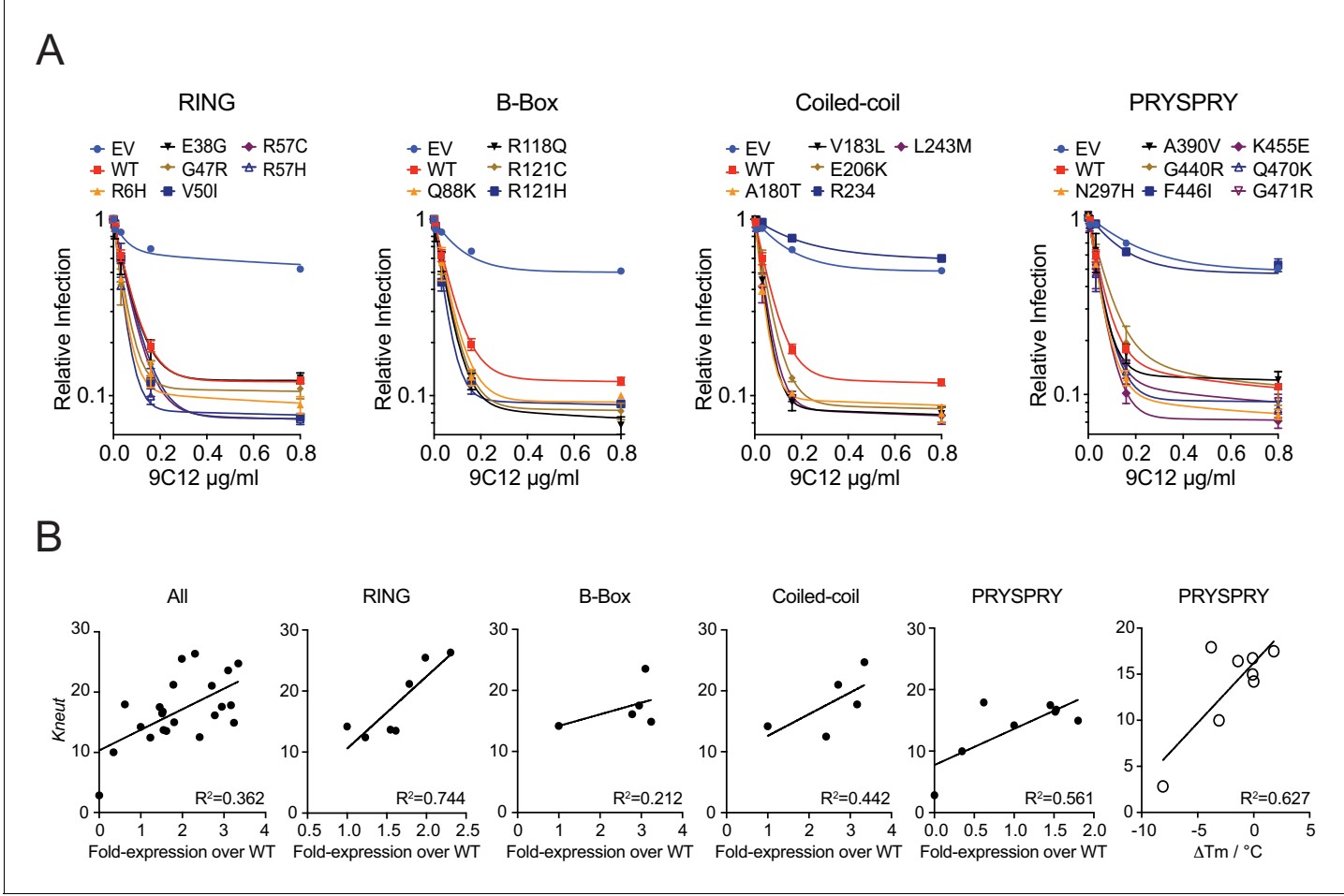

**Figure 6.** Viral neutralization by TRIM21 natural variants. (A) Neutralization experiments were carried out in *TRIM21* KO 293T cell lines stably reconstituted with TRIM21 variants expressed at endogenous levels. Each stable cell line was challenged with AdV5-GFP in the presence of the anti-hexon monoclonal antibody 9C12. The AdV5 vector contains a copy of the GFP gene and relative infection levels were quantified by flow cytometry and normalized to that of virus only condition. Data compiled from at least two independent experiments (mean ± SEM) and fitted to a one phase exponential decay. (B) Correlation of neutralization efficiency ($K_{neut}$, the exponential decay constant calculated from (A)), with cellular protein expression levels (from *Figure 5*) or thermostability ($\Delta Tm$) using linear regression analysis in GraphPad Prism7. Variants are grouped into their host domains. The R234* variant was excluded from correlative analysis. Data provided in *Figure 6—source data 1*.

DOI: https://doi.org/10.7554/eLife.48339.015

The following source data and figure supplement are available for figure 6:

**Source data 1.**
DOI: https://doi.org/10.7554/eLife.48339.017
**Figure supplement 1.** F446I but not R234* can mediate viral neutralization with IFN-α priming.
DOI: https://doi.org/10.7554/eLife.48339.016

preserved at least wild-type neutralization activity, and some had slightly enhanced activity. Plotting expression levels of all variants against $K_{neut}$ suggests that while differences in expression impact activity they do not explain all the variation ($R^2 = 0.36$) (*Figure 6B*). The relationship between stability and activity is most clearly seen with PRYSPRY variants G440R and F446I, which have the lowest expression and also the least activity. Given that F446I maintains wild-type levels of binding to IgG, it is likely that the activity defect of this variant is entirely due to its lack of stability. Consistent with this, cells expressing the F446I variant show some neutralization activity when expression is increased by IFN-α pre-treatment (*Figure 6—figure supplement 1*). Indeed, intrinsic protein stability (Tm) is a reasonable predictor of neutralization activity, with an $R^2$ of 0.63 for PRYSPRY variants (*Figure 6B*). An outlier to this plot is variant A390V, which is significantly more active than its stability would predict. However, A390V is also the variant with the 2nd highest affinity to IgG, with a > 5 fold improvement over wild-type. It is therefore possible that the reduced stability of A390V is compensated to some degree by its improved IgG binding to maintain activity. Removing A390V improves the correlation between intrinsic stability and activity to give an $R^2$ of 0.91.

The reduced thermostability of the PRYSPRY variants A390V, G440R and F446I might mean that these variants can function better at the lower temperatures maintained in the respiratory epithelium, which is cooled by inspired air to a few degrees below the core temperature and is a primary site for adenovirus infection (*Keck et al., 2000*). Conversely, these variants may lose their activity at higher temperatures such as during a fever when the core temperature is increased. Therefore, we tested the ability of these three PRYSPRY variants to mediate virus neutralization at four different temperatures (33℃, 35℃, 37℃ and 39.5℃). The result showed that all three PRYSPRY variants functioned better at lower temperatures, particularly at 33℃, where even the F446I variant, which was non-functional at 37℃, showed some neutralization activity (*Figure 7A*). Variants A390V and G440R both showed slightly more neutralization activity than WT at 33℃ (*Figure 7A*), possibly due to their higher binding affinity for IgG Fc as measured by ITC (*Figure 3*). Neutralization activity, including that of the WT, declined as the incubation temperature was raised. However, this was much more pronounced in the less thermostable variants. Variant F446I was largely inactive even at 35℃, while A390V and G440R moved from being more active than WT at 33℃ to substantially less active at 39.5℃ (*Figure 7A*). These changes in neutralization activity were not accompanied by significant differences in protein expression level at the time of infection. Immunoblot analysis revealed no obvious changes in overall TRIM21 protein expression in cells after 24 hr incubation at the different temperatures (*Figure 7B*). This suggest that the differences in neutralization activity were due to stabilization of protein fold at lower temperatures and protein denaturation at higher temperatures. Taken together, the data demonstrate that reduced *in vitro* protein thermostability has a significant impact on cell function at different physiologically relevant temperatures.

In addition to mediating neutralization, TRIM21 also triggers immune signaling by activating transcription pathways such as NF-κB. We therefore tested whether any of the variants impact on signaling activity during viral infection. Consistent with the neutralization data, we observed that most variants remain active, preserving the ability to detect antibody-coated adenovirus and trigger NF-κB (*Figure 8A*). Also consistent with neutralization, many variants had slightly increased activity over wild-type. The overall correlation with absolute expression level was similarly low, with an $R^2$ of 0.25 (*Figure 8B*) but correlation within the PRYSPRY was good ($R^2 = 0.97$), suggesting that stability and expression of variants can impact signaling capability. Even A390V showed reduced activity, consistent with previous findings that signaling is more sensitive to perturbation than neutralization (*Foss et al., 2016*).

While our native expression system recreates endogenous regulation and protein levels, we sought independent confirmation of the results obtained using this approach. An advantage of the 1000 Genomes Project over other, larger, sequence collections is that immortalized lymphoblastoid cell lines (LCLs) were derived from every volunteer. We therefore obtained lines from each of the 22 volunteers with an annotated missense variant. We first sequenced the cell lines and confirmed that each variant is indeed present and heterozygous (*Figure 9—figure supplement 1*). Next, we chose several variants from our ectopic expression study – including the two B-Box variants R118Q and R121C, premature stop R234* and unstable PRYSPRY variant F446I and investigated their neutralization activity. This allowed assessment of each variant in their naturally occurring genomic background. Despite lower TRIM21 expression than in a comparison wild-type LCL (*Figure 9A*), each variant LCL displayed remarkably similar neutralization activity (*Figure 9E*). As the ectopic expression

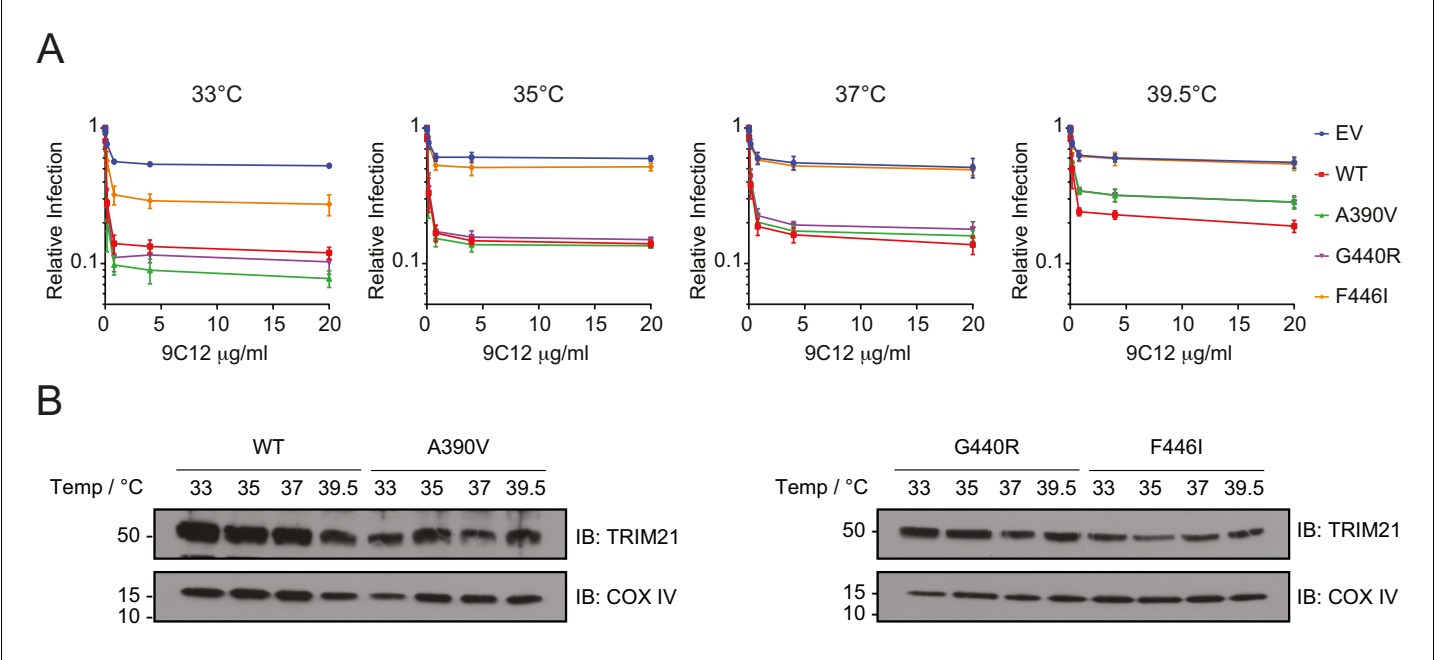

**Figure 7.** Unstable TRIM21 PRYSPRY variants function better at 33°C but lost more activity at 39.5°C. (**A**) Neutralization of AdV5-GFP in reconstituted 293T cell lines in the presence of anti-hexon 9C12 IgG at the indicated incubation temperatures. Relative infection levels were quantified by flow cytometry and normalized that of virus only condition. Data compiled from two independent experiments (mean ± SEM). (**B**) Immunoblot of TRIM21 and COX IV (loading control) in lentivector reconstituted 293T cells expressing the indicated TRIM21 variant after 24 hr incubation at the indicated temperatures. Data provided in *Figure 7—source data 1*.

DOI: https://doi.org/10.7554/eLife.48339.018

The following source data is available for figure 7:

**Source data 1.**

DOI: https://doi.org/10.7554/eLife.48339.019

data shows that both R234* and F446I variants are non-functional for neutralization as homozygotes (*Figure 6A*), this suggests that the remaining wild-type copy of TRIM21 is sufficient to maintain full activity. The data also indicates that non-functional TRIM21 variants do not exert a dominant negative effect, despite the protein being dimeric. To further test this, we established stable lines expressing variants R234* and F446I in wild-type 293T cells that retain endogenous TRIM21 expression (*Figure 9B*). We observed no reduction in either neutralization (*Figure 9F*) or signaling (*Figure 9I*) capability in these cells, consistent with the LCL data. Together this suggests that deleterious TRIM21 mutants will be autosomal recessive.

Volunteer LCLs allow functional testing of variants in situ but the available haplotypes in our study are all heterozygous. We therefore chose two variants – B-Box mutant R118Q and premature stop R234* – and carried out CRISPR gene editing to further confirm the validity of our ectopic expression approach. We isolated both heterozygous and homozygous gene edits of each variant (*Figure 9C–D*) and tested their respective neutralization (*Figure 9G*) and signaling activities (*Figure 9J*). For the B-Box variant R118Q, we found that heterozygous and homozygous clones were equally functional in both neutralization and signaling assays. This is consistent with results from both ectopic homozygous and volunteer LCL heterozygous experiments. For the truncated variant R234*, the heterozygous CRISPR clone maintained wild-type neutralization activity in agreement with the volunteer LCL data. Heterozygous R234* also maintained signaling activity, albeit at approximately half that of wild-type, which may be a result of reduced overall TRIM21 expression. Homozygous R234* clones were completely non-functional in both neutralization and signaling, in agreement with ectopic expression data.

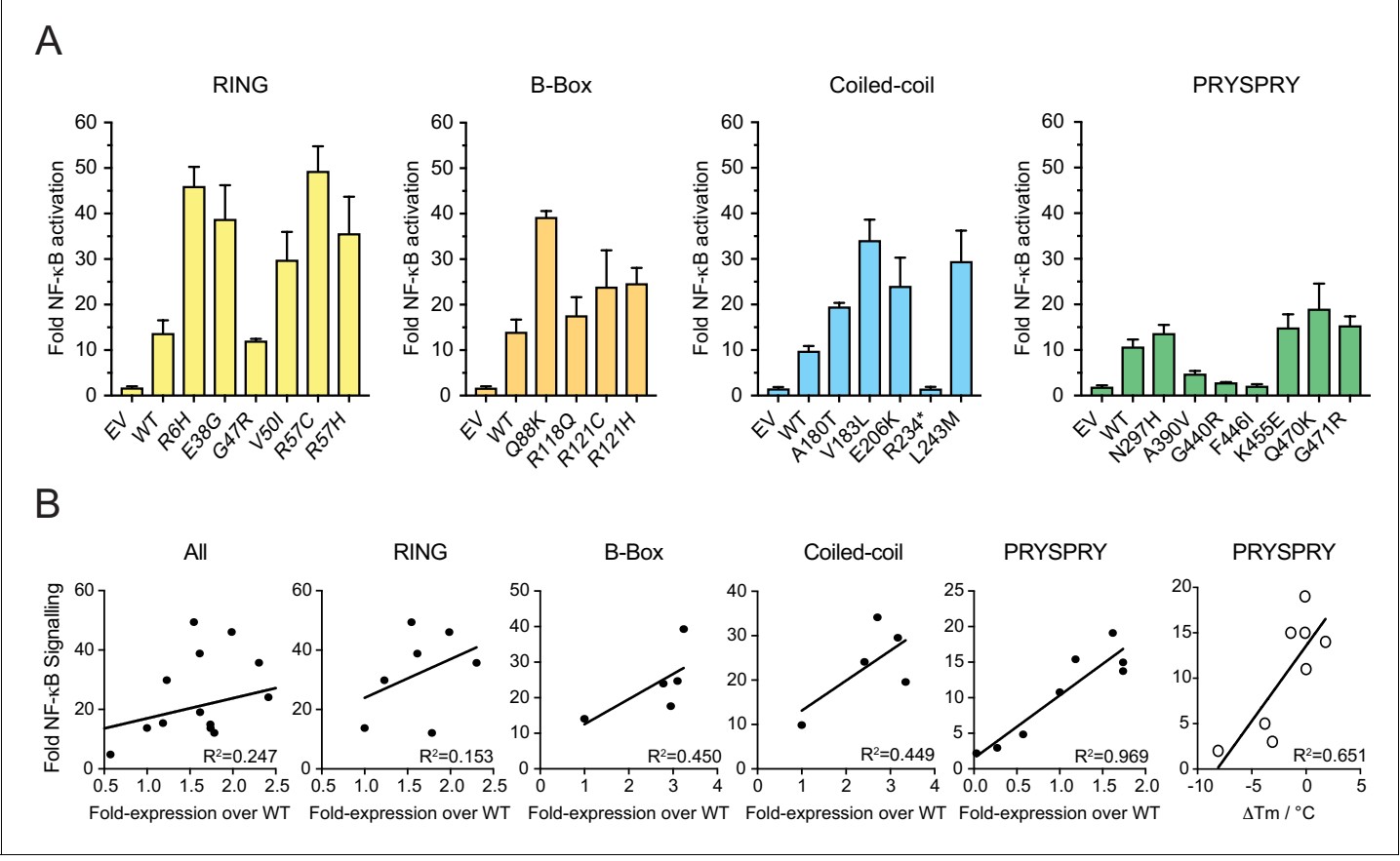

**Figure 8.** Viral sensing of by natural TRIM21 variants. (**A**) Stable 293T cell lines expressing TRIM21 variants were infected with AdV5 in the presence of anti-hexon 9C12 antibody and immune activation was measured 6 hr post infection using an NF-κB luciferase reporter. Data compiled from at least two independent experiments and expressed as fold change over that of virus only condition. EV (empty vector); mean ± SEM. (**B**) Correlation between NF-κB induction and cellular expression levels or thermostability by domain using linear regression analysis in GraphPad Prism7. The R234* variant was excluded from correlative analysis. Data provided in *Figure 8—source data 1*.

DOI: https://doi.org/10.7554/eLife.48339.020

The following source data is available for figure 8:

**Source data 1.**

DOI: https://doi.org/10.7554/eLife.48339.021

## Discussion

While individually rare, missense variants are the most common form of coding diversity in the human genome. Predicting the effects of rare variants is therefore of considerable importance to human health. However, specific rare variants are difficult, if not impossible, to associate with disease in GWAS-type studies precisely because of their rarity and thus there are very few studies of the properties of disease-causing rare mutants in human proteins (*Yue et al., 2005*). We have used TRIM21 as a case study both to assess the impact of naturally occurring rare variants and test the accuracy of *de novo* prediction. Previous reports have noted a tendency for over-prediction, with anticipated defective variants showing no reduction in wild-type activity. An *in vivo* study of missense mutations in immune genes concluded that predicted false-positive deleterious variants might be due to nearly neutral mutations that do not impact function sufficiently to give a measurable phenotype (*Miosge et al., 2015*). In our study, we were able to quantify differences in all tested variants suggesting that our assays are sufficiently sensitive.

We observed that while there was some evidence of false-positives, (e.g. with SIFT and PolyPhen-2), there was remarkably accurate assessment of variant phenotypes with CADD, REVEL and MetaLR – all of which correctly predicted only F446I as a loss-of-function variant.

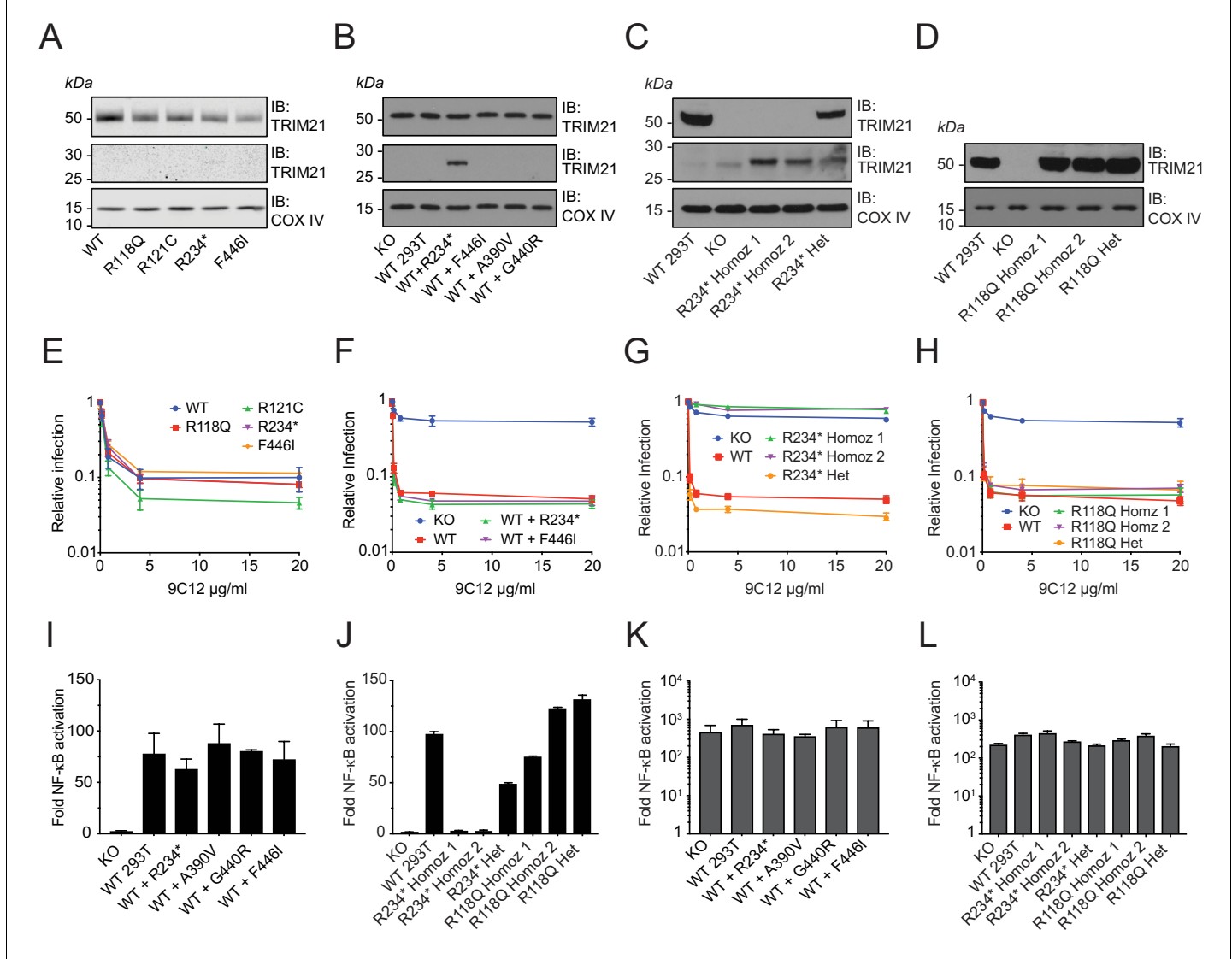

**Figure 9.** Natural missense variants do not exert dominant negative effect. (A–D) Immunoblot (IB) for TRIM21 and COX IV (loading control) in (A) selected LCLs (B) Transduced WT 293T cell lines stably expressing TRIM21 variants under the native *TRIM21* promoter. (C–D) CRISPR gene-edited 293T clones expressing the R234* and R118Q variant respectively. (E–H) AdV neutralization in the presence of anti-adenovirus hexon monoclonal IgG 9C12 in (E) selected LCLs. (F) WT 293T cells expressing the R234* and F446I variant. (G–H) CRISPR gene-edited 293T clones. Relative infection levels quantified by flow cytometry and normalized to virus only condition. Data compiled from at least two independent experiments (mean ± SEM). (I–L) Activation of NF-κB signaling by 9C12 coated AdV5 (I–J) or human TNF-α (K–L) in the respective cell lines. Data compiled from two independent experiments (mean ± SEM). Data provided in *Figure 9—source data 1*.

DOI: https://doi.org/10.7554/eLife.48339.022

The following source data and figure supplement are available for figure 9:

**Source data 1.**

DOI: https://doi.org/10.7554/eLife.48339.024

**Figure supplement 1.** Sanger sequencing chromatograms of the *TRIM21* gene in LCLs.

DOI: https://doi.org/10.7554/eLife.48339.023

We observed that overprediction is partly alleviated with increasing size of the underlying sequence database. With fewer genomes, some positions might appear conserved not because mutations at these positions are deleterious but because available datasets are too small for all possible allowed substitutions to be sampled. The improvement with increasing number of genomes

suggests taxon sampling is an important factor in improving the accuracy of predictions and suggests that it will further improve in the future as more whole genome sequences become available.

Li et al., recently performed a benchmark of predictors using recently characterized variants of clinical interest and all possible single substitutions in PPARG (*Li et al., 2018*). Our findings are consistent with their conclusions, in that REVEL, CADD and, MetaLR give the most accurate predictions while SIFT, PolyPhen-2 and MutationAssessor perform less well. Li et al., treated each mutation as binary (benign or pathogenic) but the scores reported by predictors are continuous. Detailed phenotyping based on known functions of TRIM21 that we performed here have allowed us to correlate predictions with specific types of disruptions to protein structure and function, and we noted high correlation between pathogenicity predictions and impact on protein stability. Remarkably, the best performing predictors not only accurately identify the most destabilizing mutations but also are able to correctly rank all variants. If this trend generalizes to other globular proteins, pathogenicity predictors could potentially be used to model the effect of mutations on stability, especially considering that computational methods explicitly designed for the latter perform poorly (*Thiltgen and Goldstein, 2012*).

The fact that many variants do not impact TRIM21 activity is consistent with the concept of protein robustness and the buffering of function in the presence of destabilizing mutation. Under the threshold model of robustness, destabilizing mutations retain wild-type activity while stability remains above a certain level (*Tokuriki and Tawfik, 2009*). Importantly, we observed more accurate prediction of deleterious variants in the PRYSPRY, a stable globular domain, than in the RING or B-Box – both of which are Zn-finger motifs. PRYSPRY variants A390V, G440R and F446I were predicted to be impactful and had measurably reduced antiviral activity. Meanwhile, RING variant G47R and B-Box variant R121C, although predicted by some algorithms to be deleterious, possessed wild-type function. The reduced prediction accuracy for the RING and B-Box is consistent with the notion of gradient robustness, whereby dynamic or partly disordered proteins with fewer stabilizing interactions have less to lose, in contrast to highly ordered and well-packed structures like the PRYSPRY that are poised to undergo a steep functional decline once their stability threshold is compromised. Domain architectures such as the Zn-finger, where stability is provided by an inorganic cofactor, may represent a third category that has both threshold and gradient behaviour. Mutations in chelating residues would be predicted to be individually catastrophic (threshold robustness) but elsewhere display little effect (gradient robustness). Categorizing protein folds and processing them independently may improve predictions of variant stability. If pathogenicity predictors are more effective at capturing the impact on protein stability rather than function, this may explain the poor correlation observed between observed and predicted affects in regions other than globular domains.

The accuracy of deleterious prediction is complicated by the fact that a damaging mutation cannot be equated with a disease phenotype (*Eilbeck et al., 2017*). A limitation of this study is that we have only analyzed individual SNPs and not addressed the potential for epistatic effects and co-variance of multiple SNPs either within *TRIM21* or in co-factor alleles. In part this is because there are no identified naturally occurring *TRIM21* haplotypes with multiple SNPs. We also did not find evidence of epistasis between *TRIM21* and its cognate E2 enzymes *Ube2W* or *Ube2N*, as these were wild-type in *TRIM21* variant haplotypes from the 1000 Genomes dataset. Some haplotypes do carry variant *IGHG1* alleles, the predominant IgG subclass, (including K97R, D239E, L241M, A314G, D390N, G396). However, these IGHG1 variants are relatively conservative substitutions, classified as benign/tolerated by PolyPhen and SIFT, and include common allotypes. The A314G variant is potentially of interest as it is located relatively close to the 'HNHY' motif that is bound by the TRIM21 PRYSPRY domain. However, individuals possessing both A314G and a TRIM21 variant have a different RING sequence rather than PRYSPRY, arguing against any specific epistasis.

When predicting the relative function of variants, it is important to note that even individual SNPs can be challenging as they may have opposing effects that mitigate against a deleterious outcome. For instance, although PRYSPRY variant A390V was significantly less stable than wild-type and had lower levels of cellular expression, it bound antibody fivefold more tightly and was able to maintain wild-type neutralization activity. This is reminiscent of the 'stability-activity trade-off' noted for enzyme mutations, whereby variants that increase catalytic activity do so at a cost in global stability (*Tokuriki et al., 2008*). Such trade-offs complicate *de novo* prediction. Nevertheless, our results support the hypothesis that loss of protein stability is both a major cause of disease-causing mutation and a useful correlate (*Yue et al., 2005*). The intrinsic instability of TRIM21 variants was

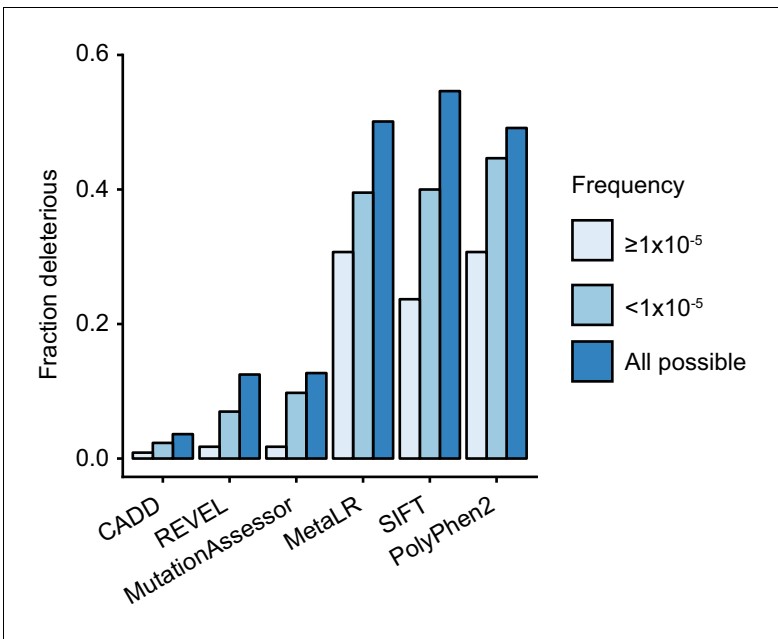

**Figure 10.** Natural rare variants are less deleterious than expected by chance. The fraction of deleterious variants calculated using the indicated algorithms for all possible variants and naturally occurring variants that are present at a frequency of $\geq 10^{-5}$ or $<10^{-5}$.

DOI: https://doi.org/10.7554/eLife.48339.025
The following source data is available for figure 10:

**Source data 1.**
DOI: https://doi.org/10.7554/eLife.48339.026

predicted with some accuracy and gave a good indicator of activity. Thermostability was also a good predictor of temperature dependent function. Of those variants with reduced thermostability, F446I, which was non-functional at 37°C, possessed some neutralization activity at 33°C, whilst A390V and G440R lost more activity than WT at 39.5°C despite their higher affinity for antibody. These results highlight the importance of considering physiological temperature when assessing the impact of altered thermostability on variant function. For instance, as is the case here, function in the respiratory tract can take place at temperatures a few degrees cooler than core body but also at temperatures above 37°C during fever.

TRIM21 antibody receptor function is maintained in most rare missense variants, suggesting that this activity is under purifying selection and therefore important in humans. In support of this, comparing the predictions of more frequent variants (frequency $\geq 1 \times 10^{-5}$) with rare variants (frequency $<1 \times 10^{-5}$) shows that more abundant variants are less likely to be deleterious (*Figure 10*). Comparison with predictions for random variants also shows that both sets of naturally occurring variants are less deleterious than those arising by chance (*Figure 10*). The absence of haploinsufficiency, toxic gain-of-function or dominant negative effects in tested natural variants means that any variant that is deleterious will be autosomal recessive. In gnomAD, a large dataset of variation in over 140,000 individuals, there is no evidence for homozygous loss-of-function variants in TRIM21. As the highest frequency of a high-confidence loss-of-function variant is $3.19 \times 10^{-5}$, likely fewer than 1 in 982 million individuals are born with no TRIM21 antibody receptor activity. This result suggests it will not be possible to infer the importance of TRIM21 in human biology based on statistical association with common diseases (e.g. as in GWAS).

# Materials and methods

## Key resources table

| Reagent type (species) or resource | Designation | Source or reference | Identifiers | Additional information |
|---|---|---|---|---|
| Gene (*Homo sapiens*) | *TRIM21* | HGNC | HGNC: 11312 | |
| Strain, strain background (*Escherichia coli*) | C41(DE3) | Sigma-Aldrich | Cat#: CMC0017 | Chemically competent cells |
| Strain, strain background (Human adenovirus type 5) | AdV5-GFP | Viraquest | N/A | |
| Strain, strain background (Human adenovirus type 5/35 chimera) | AdV5/35-GFP | Viraquest | N/A | |
| Cell line (*Homo sapiens*) | 293T | ATCC | Cat#: CRL-3216; RRID:CVCL_0063 | |
| Cell line (*Homo sapiens*) | *TRIM21* Knockout 293T | *Dickson et al., 2018* | DOI: 10.7554/eLife.32660 | CRISPR/Cas9 gene knockout |
| Cell line (*Homo sapiens*) | *TRIM21* Knockout HeLa | *Bottermann et al., 2019* | DOI: 10.1016/j.chom.2019.02.016 | CRISPR/Cas9 gene knockout |
| Cell line (*Homo sapiens*) | Lymphoblastoid cell lines | NHGRI Repository at Coriell Institute for Medical Research | RRID:SCR_004528 | |
| Recombinant DNA reagent | *pGL4.32* | Promega | Cat#: E8491 | NF-kB-RE-firefly-luciferase reporter construct (*Photinus pyralis*) |
| Recombinant DNA reagent | pCR/V1 (HIV-1 Gag-Pol) | *Zennou et al., 2004* | DOI: 10.1128/JVI.78.21.12058–12061.2004 | Lentivirus packaging vector |
| Recombinant DNA reagent | pMD2.G | Addgene | RRID:Addgene_12259 | VSV-G *env* for making pseudotyped lentiviral vector |
| Recombinant DNA reagent | pHR' | *Demaison et al., 2002* | DOI: 10.1089/10430340252898984 | HIV-1 based lentiviral vector |
| Antibody | Human plasma IgG Fc | Athens Research and Technology | Cat#: 16-16-090707-FC; RRID:AB_575814 | |
| Antibody | Anti-Human TRIM1 (Mouse monoclonal) | Santa Cruz Biotechnology | Cat#: sc-25351; RRID:AB_628286 | Immunoblot (1:1000) |
| Antibody | Anti-Human COX IV (Rabbit monoclonal) | LI-COR | Cat#: 926–42214; RRID:AB_2783000 | Immunoblot (1:5000) |
| Antibody | Anti-Mouse-HRP (Goat polyclonal) | Sigma | Cat#: A0168; RRID:AB_257867 | Immunoblot (1:5000) |
| Antibody | Anti-Rabbit-HRP (Goat monoclonal) | Sigma | Cat#: A0545; RRID:AB_257896 | Immunoblot (1:5000) |
| Antibody | Anti-mCherry (Rabbit polyclonal) | Abcam | Cat#: ab167453; RRID:AB_2571870 | IF (1:500) |
| Antibody | Anti-Human IgG Alexa Fluor 488 (Goat polyclonal) | ThermoFisher | Cat#: A-11013; RRID:AB_2534080 | IF (1:500) |
| Antibody | Anti-Rabbit IgG Alexa Fluor 568 (Goat polyclonal) | ThermoFisher | Cat#: A-11036; RRID:AB_10563566 | IF (1:500) |

*Continued on next page*

*Continued*

| Reagent type (species) or resource | Designation | Source or reference | Identifiers | Additional information |
|---|---|---|---|---|
| Antibody | IRDye 680RD Goat anti-Rabbit IgG | LI-COR | Cat#: 926–68071; RRID:AB_10956166 | Immunoblot (1:5000) |
| Antibody | IRDye 800CW Goat anti-Mouse IgG | LI-COR | Cat#: 926–32210; RRID:AB_621842 | Immunoblot (1:5000) |
| Antibody | Humanized anti-AdV5 hexon monoclonal IgG | *Foss et al., 2016* | DOI: 10.4049/jimmunol.1502601 | |
| Peptide, recombinant protein | Human TNF-α | PeproTech | Cat#: 300-01A | 10 ng/ml |
| Peptide, recombinant protein | Human IFN-α1 | Sigma | Cat#: SRP4596 | 1000U/ml |
| Sequence-based reagent | Alt-R CRISPR-Cas9 tracrRNA | IDT | Cat#: 1072532 | |
| Sequence-based reagent | Alt-R CRISPR-Cas9 crRNA (*TRIM21* KO) | IDT | N/A | ATGCTCACAGGCTCCACGAA |
| Sequence-based reagent | Alt-R CRISPR-Cas9 crRNA (*TRIM21* R118) | IDT | N/A | GTCACGGTGTTTCCGAGACT |
| Sequence-based reagent | Alt-R CRISPR-Cas9 crRNA (*TRIM21* R234) | IDT | N/A | TCATCTCAGAGCTAGATCGA |
| Sequence-based reagent | ssODN template for HDR (TRIM21 R118Q) | IDT | N/A | CACAGGGGGAACGGTGTGCAGTGC ATGGAGAGAGACTTCACCTGTTCTGT GAGAAAGATGGGAAGGCCCTTTGC TGGGTATGTGCTCAGTCTCAGAAACAC CGTGACCACGCCATGGTCCCTCTTGA |
| Sequence-based reagent | ssODN template for HDR (TRIM21 R234*) | IDT | N/A | AGCCAGGCCCTACAGG AGCTCATCTCAG AGCTAGATTGAAGATGCCACAG CTCAGCACTGGAACTGCTGCAGGT GAGACAGGGAGGGGTTTCCTTCTAC AATTCAGGGAATAACT GAAAAAGACCAG |
| Sequence-based reagent | Forward primer for *TRIM21* promoter cloning | Sigma | N/A | ggatcgataagcttgatatcgaa ttcGCATGTTG TGCACA |
| Sequence-based reagent | Reverse primer for *TRIM21* promoter cloning | Sigma | N/A | tgcttaacgcgTGTCAAGT GTGCCGTTAAACAG |
| Commercial assay or kit | Gibson Assembly Master Mix | NEB | Cat#: E2611L | |
| Commercial assay or kit | QIAprep Spin Miniprep Kit | QIAGEN | Cat#: 27106 | |
| Commercial assay or kit | QIAquick PCR Purification Kit | QIAGEN | Cat#: 28104 | |
| Commercial assay or kit | QIAquick Gel Extraction Kit | QIAGEN | Cat#: 28704 | |
| Commercial assay or kit | Neon Transfection System 10 µL Kit | Invitrogen | Cat#: MPK1096 | |
| Software, algorithms | GraphPad Prism 7 | GraphPad | RRID:SCR_002798 | |
| Software, algorithms | R | R Project for Statistical Computing | RRID:SCR_001905 | |

*Continued on next page*

*Continued*

| Reagent type (species) or resource | Designation | Source or reference | Identifiers | Additional information |
|---|---|---|---|---|
| Other | Hoechst 33342 Solution | Thermo Scientific | Cat. #: 62249 | (1 μg/mL) |
| Other | Steadylite plus Reporter Gene Assay System | PerkinElmer | Cat. #: 6066751 | |
| Other | Fugene 6 Transfection Reagent | Promega | Cat. #: E2691 | |
| Other | Genome Aggregation Database | Broad Institute | RRID:SCR_014964 | |
| Other | 1000 Genomes Project | 1000 Genomes Project | RRID:SCR_006828 | |

## Plasmids

Plasmids pOPT-6His-PRYSPRY was generated in a previous study (*James et al., 2007*). Point mutations were generated using the modified QuickChange protocol (*Liu and Naismith, 2008*). pGL4.32 containing a firefly luciferase cassette under NF-κB response elements was obtained from Promega. pHR' was originally from Dr Adrian Thrasher. pMD2.G encoding VSV-G was obtained from Addgene and pC/RV1 encoding HXB2 derived HIV-1 Gag-pol was produced by *Zennou et al. (2004)*. All plasmids were grown in *E. coli* DH10B strain and purified using QIAprep Spin Miniprep Kit (QIAGEN).

## Antibodies

Pooled human serum IgG Fc fragment was obtained from Athens Research and Technology. Mouse anti-AdV5 hexon monoclonal antibody 9C12 and its humanized versions were produced and characterized in previous studies (*Foss et al., 2016*; *Bottermann et al., 2016*). Antibodies used for immunoblotting were anti-TRIM21 (52 kDa Ro/SSA Antibody D-12 Santa Cruz Biotechnology, 1:500), anti-COX IV (LI-COR, 926–42214, 1:5000), anti-mouse-HRP (Sigma, A0168, 1:2500) and anti-Rabbit-HRP (Cell Signaling, 7074, 1:5,000), IRDye 800CW Goat anti-Mouse IgG (LI-COR Bioscience, 925–32210, 1:5000), IRDye 680RD Goat anti-Rabbit IgG (LI-COR Bioscience, 925–68071, 1:5000). For immunofluorescence, the primary antibody used was rabbit anti-mCherry polyclonal antibody (Abcam, ab167453 diluted 1:500). The secondary antibodies used were Alexa Fluor 488-labeled goat anti-human and Alexa Fluor 568-labeled goat anti-rabbit anibodies (Thermofisher) diluted 1:500.

## Cells

Human embryonic Kidney 293T cells (293Ts) and HeLa cells were maintained in Dulbecco's Modified Eagle Medium (DMEM) supplemented with 10% (v/v) of fetal bovine serum (FBS), 100 U/ml penicillin and 100 μg/ml streptomycin. Lymphoblastic cell lines (LCLs) were obtained from Coriell Biorepositories and were maintained in RPMI-1640 media supplemented with 15% (v/v) of FBS, 100 U/ml penicillin and 100 μg/ml streptomycin. hTERT-RPE1 cells were maintained in DMEM/F-12 supplemented with 10% (v/v) of FBS, 100 U/ml penicillin and 100 μg/ml streptomycin. Unless otherwise stated, cells were incubated at 37°C with 5% $CO_2$ and passaged every 3–4 days.

*TRIM21* knockout (K.O) 293T and HeLa cell lines were generated using the CRISPR-Cas9 gene editing system as described previously (*Dickson et al., 2018*; *Bottermann et al., 2019*).

## Interferon

For interferon treatment, recombinant human interferon-alpha (Sigma) was added to the media at a final concentration of 1000 IU/ml during cell plating.

## Variation data

All TRIM21 missense variants and corresponding sample identifiers were extracted from vcf files available on the 1000 Genomes website. Missense variants from gnomAD with allele number greater

than 20,000 in TRIM21, TRIM5, TRIM20, HDACs, ACTs and HLAs, were obtained from https://gno-mad.broadinstitute.org/.

## Protein expression and purification

N-terminally His-tagged TRIM21 PRYSPRY protein and point mutants were expressed in *E. coli* C41 strain. The cells were grown in 2 X TY media with 100 µg/ml ampicillin at 37℃ until an $OD_{600}$ of 0.7 before induction with 1 mM Isopropyl β-D-1-thiogalactopyranoside (IPTG) and incubation overnight at 18℃. Cells were lysed by sonication in buffer containing 50 mM Tris at pH 8, 150 mM NaCl, 5 mM imidazole, 2 mM Dithiothreitol (DTT), 20% (v/v) BugBuster (Novagen) and 1 X cOmplete protease inhibitor (Roche). Lysates were cleared by centrifugation at 37,000 x g, 4℃ for 1 hr. Purification of N-terminally His-tagged TRIM21 PRYSPRY proteins were performed as described (*James et al., 2007*).

## Isothermal Titration Calorimetry (ITC)

Samples were dialyzed overnight at 4℃ into 5 mM ammonium acetate and 200 mM NaCl buffer. Protein concentrations were measured using a Nanodrop 2000 (Thermo Scientific) with extinction coefficients calculated from Protpram (*Gasteiger E et al., 2005*). ITC measurements were performed using a MicroCal iTC200 (Malvern) with 40 µM IgG Fc in the syringe and 8 µM of TRIM21 PRYSPRY in the cell unless otherwise stated. The experiments were conducted at 20℃ with 16–20 injections of 2–2.5 µl. For data analysis, the heat of dilution was subtracted from the raw data and the result was fitted using a single class binding site model in the manufacturer's software to determine the dissociation constant ($K_d$) and stoichiometry of binding (N).

## Nano differential scanning fluorimetry (NanoDSF)

Protein samples were diluted to 2 µM in buffer containing 50 mM Tris-HCl pH 8.0, 150 mM NaCl, 1 mM DTT. 10 µl of diluted samples were loaded into Prometheus NT.48 Series nanoDSF Grade Standard Capillaries (NanoTemper Technologies). Melt curves were obtained using the Prometheus NT.48 (NanoTemper Technologies) instrument by heating protein samples from 20℃ to 95℃ with a temperature slope of 2℃/min.

## Generation of native TRIM21 promoter-driven lentiviral vector

The pHR' vector (originally from Dr. Adrian Thrasher) was linearized using EcoRI-HF (NEB) and NotI-HF (NEB) restriction enzymes to remove the original promoter sequence. The human *TRIM21* 2 kb upstream promoter sequence was amplified by PCR from genomic DNA extracted from 293T cells using primers containing overlap with the linearized lentivector plasmid backbone and 5' of *TRIM21* coding sequence. Variant *TRIM21* coding sequences were amplified with overhangs for both the promoter sequence and the linearized lentivector plasmid backbone. The PCR products and the linearized lentivector plasmid were purified by gel extraction using the QIAquick Gel Extraction Kit (QIAGEN) or QIAquick PCR purification Kit (QIAGEN). The inserts were ligated into the lentivector backbone by Gibson assembly (NEB) whilst preserving the original restriction sites.

## Lentiviral vector production and cell transduction

Lentivirus vector were produced by co-transfecting $5 \times 10^6$ WT 293Ts in 10 cm dishes with 2 µg pCRV-GagPol, 1 µg pMD2G-VSVg and 2 µg pNatP-TRIM21 using Fugene-6 transfection reagent (Promega). The media were changed after 24 hr and virus containing supernatant were harvested and filtered (0.45 µm syringe filter) after 48 hr. For transduction, 293T cells were plated in six-well plate at $1 \times 10^5$ cells/well a day before. 293T cells were infected with equal volumes of lentivirus containing supernatants in the presence of 5 µg/ml polybrene. 48 hr after transduction, the transduced cells were subject to selection using puromycin at a final concentration of 2.5 µg/ml.

## Immunoblotting

Whole cell lysates were prepared using 25 µl RIPA buffer (Sigma-Aldrich) per $1 \times 10^6$ cells following manufacture's protocol. The cell lysates were boiled at 98℃ for 5 min in 1 X NuPAGE LDS sample buffer (Invitrogen) containing 50 mM DTT. Equal volumes of sample were loaded per lane onto NuPAGE 4–12% Bis-Tris gels (Invitrogen) and resolved in 1 X MOPS-SDS running buffer at 185 V for

55 min. Protein transfer onto nitrocellulose membrane was performed using an iBlot Gel Transfer Device (Invitrogen) as per manufacturer's instructions. All blots were blocked in 5% (w/v) non-fat milk in PBST (PBS with 0.1% Tween 20) and incubated in antibody diluted in 5% milk PBST and washed in PBST. Visualization was performed using either LI-COR Odyssey CLx Near-infrared imaging system or Amersham ECL, ECL SELECT detection reagents (GE Healthcare Life Sciences) following to the manufacturer's protocols. Band intensity were quantified using Image Studio Lite (LI-COR) software.

## Immunofluorescence

293T cells reconstituted with mCherry-TRIM21 was plated at a density of $1 \times 10^5$ cells per well in 24-well plates on poly-D-lysine-coated coverslips (Corning) with 1000 U/ml IFN-α. The following day, the cells were infected with 48 µl/well of human 9C12 IgG1 (20 µg/ml) coated AdV5 vector (ViraQuest, diluted 1:4) complex mixed 1:1 in 150 µl of serum free DMEM with cold attachment at 4°C for 30 min. The cells were fixed with 4% paraformaldehyde solution (Pierce) according to manufacturer's protocol 30 min after the cells were returned to 37°C. The fixed cells were permeabilized with 0.1% Triton X-100 in PBS for 1 hr at 20°C and blocked in 3% BSA in PBS with 0.1% Triton X-100 for 1 hr at 20°C. All antibodies and Hoechst 33342 stain (1:1000) were diluted in 3% BSA in PBS with 0.1% Triton X-100 and incubated for 1 hr at 20°C. Images were acquired on Leica SP8 equipped with HC PL APO 63x/1.40 Oil PH3 CS2 objective. Image analysis and colocalization was calculated using the ComDet plugin in Fiji (Schindelin et al., 2012).

## Virus neutralization assay

293T cells were plated at a density of $5 \times 10^4$ cells/well in 24-well plates and were allowed to attach overnight. AdV5-GFP (ViraQuest) was diluted 1:1250 in PBS and AdV5/35 chimera-GFP (ViraQuest) was diluted 1:25 in PBS. The diluted viruses were mixed 1:1 with 9C12 antibody at the indicated concentrations and incubated for 1 hr at 20°C to allow for complex formation. 10 µl of the virus or virus-antibody complex was added per well. Cells were harvested 16–20 hr postinfection and evaluated for GFP expression on a BD LSRFortessa cell analyzer (BD Biosciences). The results were analyzed using FlowJo software (FlowJo LLC) by gating for cell population using forward and side scatter; the background threshold for GFP signal was set at ~0.1% using uninfected control cells. Relative infection was calculated by normalizing to the infection level in the absence of antibody. The neutralization constant ($K_{neut}$) was calculated using the method described (McEwan et al., 2012).

## NF-κB signaling assay

293T cells were plated at a density of $3 \times 10^5$ cells/well in a 6-well plate. 24 hr later the cells were transfected with 200 ng of pGL4.32 NF-κB luciferase (Promega) using FuGENE 6 Transfection reagent (Promega). Cells were incubated for 6 hr before reseeding at a density of $1 \times 10^4$ per well in Corning CellBIND 96-well plates and allowed to adhere overnight. AdV5-GFP (ViraQuest) was diluted 1:3 in PBS and mixed 1:1 with 20 µg/ml humanized 9C12 IgG3b or PBS and incubated for 1 hr at 20°C to allow for complex formation. 5 µl of the virus or virus-antibody complex was added to each well (100 µl of media) in triplicates and allowed to incubate for 6 hr before the cells were lysed with 100 µl/well steadylite plus luciferase reagent (Perkin Elmer). The luciferase activity was quantified using a BMG PHERAstar FS plate reader.

## Generation of CRISPR Knock-in cell lines

Generation of knock-in 293T cell lines were achieved by electroporation of Cas9/gRNA ribonucleoprotein complexes (Cas9-RNP) along with single stranded DNA (ssDNA) donor template for homology directed repair (HDR). The recombinant Cas9-2NLS-GFP protein was produced as described (Jinek et al., 2012). The HDR donor templates were designed according to Richardson et al. (2016). Synthetic ssDNA donor templates, tracrRNA and crRNA against TRIM21 (TCATCTCAGAGCTAGATCGA for R234*; GTCACGGTGTTTCCGAGACT for R118Q) were obtained from IDT. tracrRNA-crRNA complexes were assembled by incubating at 95°C for 5 min followed by cooling on the benchtop to 20°C. The RNA complexes were mixed with recombinant Cas9 protein at a molar ratio of 1.2:1 and incubated at 37°C for 10 min to form Cas9 RNP complexes. 50 pmol of Cas9 RNP and ssDNA HDR donor template were introduced into $8 \times 10^5$ 293Ts using the

Neon Transfection System (Invitrogen) with a setting of 2 pulses of 1400 V for 20 ms. 48 hr post electroporation, the cells were cloned by fluorescence-activated cell sorting into 96-well plates (1 cell/well). Knock-in mutations were confirmed by PCR amplification of gene segments followed by Sanger sequencing.

## Variant effect predictions

Predictions for 1000 Genomes variants for SIFT, PolyPhen-2, CADD, REVEL, MetaLR and MutationAssessor were obtained from Ensembl (*Cunningham et al., 2019*). CADD, REVEL, MetaLR and MutationAssessor predictions for amino-acid mutations resulting from all possible single-nucleotide substitutions were obtained from dbNSFP (*Liu et al., 2013*). Predictions for the same set of mutations for SIFT and PolyPhen-2 (based on the UniRef 2014_11 in the case of SIFT and UniProtKB 2013_10 in the case of PolyPhen-2) were obtained from Ensembl using the Perl API. Predictions for older versions of the underlying databases were obtained from http://sift.bii.a-star.edu.sg/ using UniProtKB2010_09 in the case of SIFT and from http://genetics.bwh.harvard.edu/pph2/ using UniProtKB2012_01 in the case of PolyPhen-2. Predictions were classified as deleterious if they were predicted as following the Ensembl guidelines. In the case of PolyPhen-2, 'possibly deleterious' and 'probably deleterious' predictions were merged into one category.

## Acknowledgements

The authors thank the Genome Aggregation Database (gnomAD) and the groups that provided exome and genome variant data to this resource. A full list of contributing groups can be found at http://gnomad.broadinstitute.org/about. We thank Dr Claire Dickson, Dr Donna Mallery, Dr Adam Fletcher, Dr William McEwan, Dr Chris Johnson, Dr Maria Bottermann, Dr Dean Clift, Dr Larisa Labzin, Dr Lorena Boquete Vilarino and Dr Marina Vaysburd for valuable discussions. We also thank Dr Jan Terje Andersson for the kind gift of humanized 9C12 antibodies and Joanne Westmoreland for the illustration of the lentivector map. The Flow cytometry data were generated with support from the Flow Cytometry facility, MRC LMB. We also thank the biophysics and light microscopy facilities at the MRC LMB for their support in this study.

## Additional information

### Funding

| Funder | Grant reference number | Author |
|---|---|---|
| Medical Research Council | U105181010 | Jingwei Zeng Greg Slodkowicz Leo C James |
| Wellcome | 200594/Z/16/Z | Leo C James |
| Rosetrees Trust | | Jingwei Zeng |
| University of Cambridge | | Jingwei Zeng |

The funders had no role in study design, data collection and interpretation, or the decision to submit the work for publication.

### Author contributions

Jingwei Zeng, Conceptualization, Data curation, Formal analysis, Funding acquisition, Validation, Investigation, Visualization, Methodology, Writing—original draft, Project administration, Writing—review and editing; Greg Slodkowicz, Conceptualization, Resources, Data curation, Software, Formal analysis, Investigation, Visualization, Methodology, Writing—original draft, Project administration, Writing—review and editing; Leo C James, Conceptualization, Resources, Data curation, Software, Formal analysis, Supervision, Funding acquisition, Investigation, Visualization, Methodology, Writing—original draft, Project administration, Writing—review and editing

**Author ORCIDs**
Jingwei Zeng https://orcid.org/0000-0002-9802-6019
Greg Slodkowicz https://orcid.org/0000-0001-6918-0386
Leo C James https://orcid.org/0000-0003-2131-0334

**Decision letter and Author response**
Decision letter https://doi.org/10.7554/eLife.48339.030
Author response https://doi.org/10.7554/eLife.48339.031

## Additional files

### Supplementary files

• Supplementary file 1. Rare missense variants in TRIM21 present in the 1000 Genomes study. All variants listed have an allele frequency of <1%. The number of heterozygous and homozygous haplotypes are indicated for each variant, together with its domain location, predicted deleteriousness (using various algorithms, see main text) and measured neutralization ($K_{neut}$) and signaling activity ($K_{sig}$). $K_{neut}$ is a measure of the efficiency of TRIM21-mediated adenovirus neutralization as defined by the exponential decay constant calculated from *Figure 6A*. $K_{sig}$ is a measure of signaling ability defined by the fold-change in NF-κB reporter activity upon adenovirus infection in the presence of antibody (*Figure 8*).
DOI: https://doi.org/10.7554/eLife.48339.027

• Transparent reporting form
DOI: https://doi.org/10.7554/eLife.48339.028

### Data availability

All data generated or analysed during this study are included in the manuscript and supporting files. Previously published data from the 100 Genomes Project (2015; http://www.internationalgenome.org/data#download) and the Genome Aggregation Datatbase (2016; https://gnomad.broadinstitute.org/downloads) was used as part of this work.

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
