## [Decision Letter]

Thank you for submitting your article "Rare missense variants in the human cytosolic antibody receptor preserve antiviral function" for consideration by *eLife*. Your article has been reviewed by two peer reviewers, and the evaluation has been overseen by Wes Sundquist as the Reviewing Editor and Patricia Wittkopp as the Senior Editor. The following individuals involved in review of your submission have agreed to reveal their identity: Owen Pornillos (Reviewer #1); Alain Fischer (Reviewer #2).

The reviewers have discussed the reviews with one another and the Reviewing Editor has drafted this decision to help you prepare a revised submission.

Summary:

James and colleagues delve into the relationships between rare non-synonymous variants and protein function. In the Introduction, the authors point out that variants that are "individually scarce but collectively abundant" may account for the "missing heritability" problem that underlies efforts to genetically account for inherited diseases. The key question they seek to address is whether "predicted deleterious mutations are actually often neutral or does phenotypic characterization fail to capture their impact?" This question was addressed by investigating the cytosolic antibody receptor TRIM21, which directs antibody-coated pathogens for proteasomal degradation and activates immune signaling pathways. Multiple biochemical and cellular assays were used to quantify a variety of functional phenotypes, and the results compared with predictions from different algorithms. The phenotypic characterization is quite thorough and rigorous, which an important strength of this study. The study supports these conclusions: (1) most of the variants are phenotypically neutral; (2) loss of function best correlates with loss of protein stability; (3) prediction algorithms are fairly accurate in assessing possible phenotypes; although there is a tendency for false positives, the authors note that this is alleviated with increasing size of the underlying sequence database (and presumably, therefore, will be corrected with time). Overall, the study is an important contribution and TRIM21 emerges as an ideal model system given the array of assays that could be put to bear. The data are well presented, convincing and the paper well written, although the conclusion that empirical testing will be an important component of any rigorous effort to assess the effects of genetic variation on protein function and disease is perhaps unsurprising.

Essential revisions:

The PRYSPRY data suggest that temperature sensitivity of the mutants would be a useful parameter to consider. Testing the SPRY mutants for temperature dependence would strengthen this work, and also test whether this is a useful general strategy (important given that the paper is, in part, a model showing how analyses should be performed in other analogous cases).

---

## [Author Response]

Essential revisions:The PRYSPRY data suggest that temperature sensitivity of the mutants would be a useful parameter to consider. Testing the SPRY mutants for temperature dependence would strengthen this work, and also test whether this is a useful general strategy (important given that the paper is, in part, a model showing how analyses should be performed in other analogous cases).

We thank the reviewers for this excellent suggestion. The potential for the unstable PRYSRY mutants to demonstrate temperature dependence in cells has interesting physiological implications. The antiviral functions of TRIM21 have been well characterised in the context of human adenovirus 5, which most commonly infects the respiratory epithelium and typically causes symptoms of the common cold. It is appreciated that the temperature of the respiratory tract is a few degrees Celsius cooler than core body temperature due to the cooling effect of inspired air (Keck et al., 2000). The three PRYSPRY variants that have reduced thermostability (A390V, G440R and F446I) may function better at the lower temperatures found in the respiratory epithelium which is a common primary site of infection. Conversely, these variants may have reduced functionality in the event of a fever when core body temperature is increased. This may not be entirely detrimental to the host as it could provide a potential negative feedback mechanism that removes the TRIM21 contribution to the intense cytokine response that underlies pathogenic fever.

We tested the ability of TRIM21 PRYSPRY variants A390V, G440R and F446I to mediate antibody dependent intracellular neutralisation (ADIN) at four different temperatures (33°C, 35°C, 37°C and 39.5°C). The results showed that all three PRYSPRY variants functioned better at lower temperatures, particularly at 33°C where the F446I variant, which was non-functional at 37°C, showed some neutralisation activity. The F446I variant quickly lost its activity as the incubation temperature was increased, showing little or no activity at 35°C, which nicely correlates with its significantly reduced thermostability. Interestingly, variants A390V and G440R showed greater activity than wild-type at 33 but lesser activity at 39.5. This is consistent with the increased affinity these variants have for antibody but reduced thermostability. These changes in neutralisation activity were not accompanied by significant differences in protein expression level at the time of infection. Immunoblot analysis revealed no obvious alteration in TRIM21 protein expression in cells after 24-hour incubation (time of infection) at the different temperatures. This suggests that the observed differences in neutralisation activity were due to stabilisation of protein fold at lower temperatures and protein denaturation at higher temperatures. The transition point at which denaturation occurs is clearly different for each variant and is dependent on the intrinsic thermostability of the protein.

The results of this experiment have been added to the revised manuscript as new Figure 7 along with a description of the findings in the Results section (eleventh paragraph) and a summary in the Discussion (fifth and sixth paragraphs).